# Comparative Efficacy of *Chondrosterum purpureum* and Chemical Herbicides for Control of Resprouts in Tanoak and Bay Laurel

Simon Francis Shamoun [1],* and Marianne Elliott [2]

1   Pacific Forestry Centre, Canadian Forest Service, Natural Resources Canada, 506 West Burnside Road, Victoria, BC V8Z 1M5, Canada
2   Puyallup Research and Extension Center, Washington State University, 2606 West Pioneer, Puyallup, WA 98371, USA; melliott2@wsu.edu
*   Correspondence: simon.shamoun@nrcan-rncan.gc.ca; Tel.: +1-250-298-2358

**Abstract:** The invasive Oomycete pathogen *Phytophthora ramorum* has killed millions of susceptible oak and tanoak trees in California and southern Oregon forests and is responsible for losses in revenue to the nursery industry through mitigation activities. In addition, infestation of forests in the United Kingdom by this organism has resulted in the destruction of many hectares of larch plantations. Resprouting stumps can be a reservoir for the inoculum of *P. ramorum* persisting on a site. In areas where the application of herbicides is not permitted, a biocontrol treatment would be an indispensable alternative. Treatment of stumps with the sap-rotting fungus *Chondrostereum purpureum* (Pers.) Pouzar has been shown to be an effective tool for the suppression of resprouting on several species, most notably red alder. In this project, the ability of *C. purpureum* to suppress resprouting was evaluated on stumps of two host species, tanoak (*Notholithocarpus densiflorus*) and California bay laurel (*Umbellularia californica*). Laboratory testing of three California isolates of *C. purpureum* indicated that the fungus can colonize bay laurel stems. Field trials were established near Brookings, Oregon, on tanoak and on bay laurel near Soquel, California. Early results of field testing showed that *C. purpureum* was able to colonize the stumps of tanoak following treatment and was found to occur naturally on tanoak logs and stumps. Formulations of *C. purpureum* appear to have some effect on reducing sprout survival in tanoak, but the most effective and rapid treatment for this host is the hack and squirt method of applying the herbicide imazapyr. Sprayed herbicide prevents sprouting on bay laurel, and there was evidence that resprouting was inhibited on stumps treated with *C. purpureum*. Over time, applications of *C. purpureum* may be a more permanent solution as the stumps begin to decay.

**Keywords:** biological control; *Chondrosterum purpureum*; forest vegetation management; sprout control; *Phytophthora ramorum*; sudden oak death; wood decay fungi; California bay laurel; tanoak

## 1. Introduction

As global plant trade expands, tree disease epidemics caused by pathogen introductions are increasing. These problems are exacerbated under the present conditions of rapid climatic change [1]. Epidemics of the invasive alien pathogen *Phytophthora ramorum* Werres, De Cock & Man in't Veld in the western USA and Europe have caused significant economic losses in the nursery trade and forest industries [2]. On the west coast of the USA, *P. ramorum* causes sudden oak death disease through the development of lethal bole cankers on several *Quercus* species, especially coast live oak (*Quercus agrifolia* Née) and on tanoak (*Notholithocarpus densiflorus* (Hook. and Arn.) Manos, Cannon and S.H. Oh), and non-lethal leaf blight on many trees and shrubs including bay laurel (*Umbellularia californica* (Hook. and Arn.) Nutt.). Since its emergence in the 1990s in California and Western Europe [2], *P. ramorum* has killed millions of trees in California forests [3] and has affected woodlands

in the UK, Ireland, and northern France. In southwestern Oregon, an aggressive eradication program has included the use of chemical herbicides, cutting, and burning of tanoak in an effort to eradicate *P. ramorum* [4]. It was soon apparent that tanoak resprouts were highly susceptible to *P. ramorum* and that infected sprouts hamper eradication efforts by maintaining the pathogen inoculum on site [5]. A similar situation occurred in north coastal counties of California (Mendocino, Humboldt, and Del Norte), where isolated infection centers existed [6]. Controlling the production of susceptible sprout material will undoubtedly be an issue in the success of these eradication activities (Figure 1). Important issues regarding these treatments are concerns over the use of chemical herbicides on public lands, particularly in riparian habitats, and the costs and efficiencies associated with mechanical treatment of sprouting vegetation. Tanoak is a native species and not considered a noxious weed, but tanoak stumps need to be treated or they will persist as an inoculum source for *P. ramorum*. California bay laurel is another important inoculum source and a prolific resprouter [7,8]. Therefore, developing environmentally and economically feasible tools to effectively control sprouting other than by chemical means is needed, particularly in ecologically sensitive sites such as wildlife habitats, recreational use sites (e.g., National Parks), or stands with high cultural value. The basidiomycete *Chondrostereum purpureum* (Pers. ex Fr.) Pouzar is a white rot fungus of many broadleaf trees that occurs worldwide in temperate zones. It is a facultative saprophyte with broad-spectrum pathogenicity toward many hardwood species. A primary invader of woody angiosperms, *C. purpureum* usually enters its host through a fresh wound, cut stump, or stem lesion [9,10]. The fungus grows through xylem tissues of the host plant, causing cambial necrosis, decay, sapwood staining, and sometimes death of the host [11,12]. Infection by *C. purpureum* can also cause foliar discoloration or silvering, known as silver-leaf, of orchard fruit trees [13]. This fungus has been used as a biological control agent for the management of weedy woody vegetation in conifer plantations and utility-rights-of-way in North America, many European countries, and New Zealand [14–18].

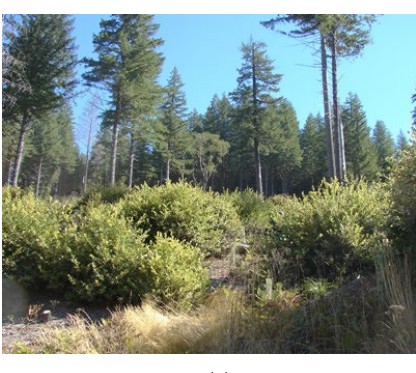 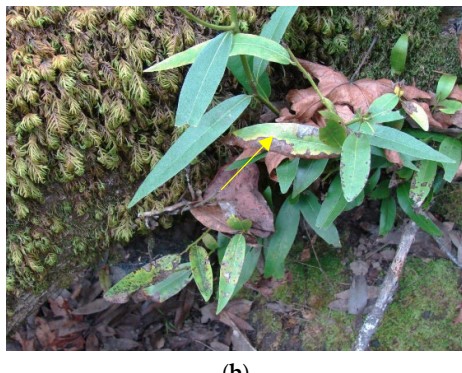

(a)  (b)

**Figure 1.** (**a**) Resprouting tanoak near Brookings, OR; (**b**) *Phytophthora ramorum* lesions on bay laurel resprouts (arrow).

The objectives of this research study were to establish field trials in southwestern Oregon (Tanoak site) and the Soquel Demonstration State Forest site in California (bay laurel site) to examine and evaluate the efficacy of the potential application of a formulated product of *C. purpureum* to inhibit the resprouts in tanoak and bay laurel stumps compared to manual brushing and chemical herbicide treatments.

## 2. Results

### 2.1. Laboratory

#### 2.1.1. Bay Laurel Isolate Selection

Stem inoculations in the laboratory and greenhouse indicated that isolate 2249 was the most aggressive on bay laurel. In the greenhouse trials, the second pass-through showed isolate 2434 and 2249 as comparable aggressors. The third and final re-isolation and pass-

through yielded a leaf turning yellow at the point of inoculation and stem discoloration from isolate 2249 (Figure 2). This suggests that isolate 2249 has adapted more to colonizing the bay laurel host tissue than isolates 2434 and 2367. On the basis of these results, we chose isolate 2249 for biocontrol formulation to be used in the California field trials.

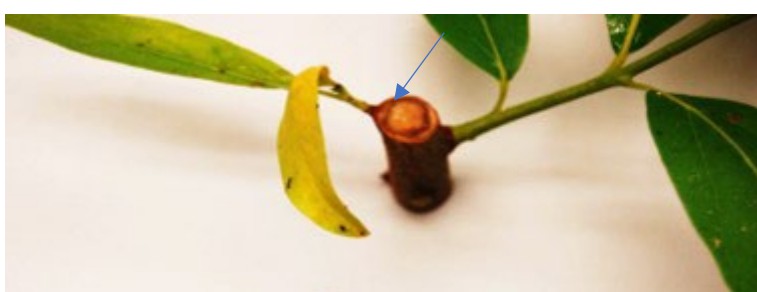

**Figure 2.** *Chondrostereum purpureum* isolate 2249 inoculated onto potted bay laurel stem in greenhouse trial. Stem discoloration is shown with an arrow.

### 2.1.2. Identification of Fungi Collected from Stumps

Fruiting bodies of fungi on decaying tanoak logs and stumps were collected in October 2010 (Figure 3). Fungi identified included *Lenzites betulina*, *Trametes versicolor*, *Stereum hirsutum*, and *Chondrostereum purpureum*. The identities of the fungi were verified by sequence analysis of the ITS rDNA. Of these fungi, *C. purpureum*, *L. betulina*, and *T. versicolor* are not reported on tanoak in the SMML Fungus–Host database [19]. Tanoak is not listed as a host for *C. purpureum*, but *C. purpureum* fruiting bodies were observed on inoculated stumps one year after treatment. The fungus was also found occurring naturally on tanoak logs and stumps at some other sites in the area.

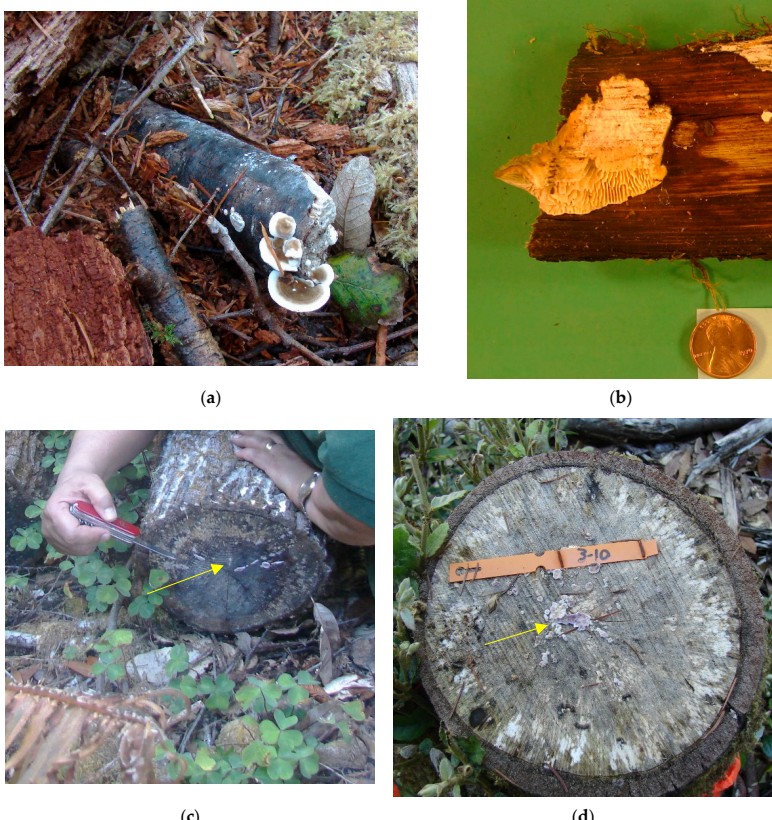

**Figure 3.** Fruiting bodies of decay fungi identified on tanoak stumps. (**a**) *Trametes versicolor*; (**b**) *Lenzites betulina*; (**c**) *Chondrostereum purpureum* (arrow) naturally occurring on tanoak log; (**d**) *Chondrostereum purpureum* (arrow) on treated stump.

Heavy deer browse was observed on most bay laurel stumps. Some browsed shoots had symptoms of *Colletotrichum* dieback. The fungus isolated from these shoots was a *Colletotrichum* spp. in the *C. acutatum* species complex, closely related to *C. clavatum* [20]. No decay or fruiting bodies of *C. purpureum* were apparent, but many stumps had evidence of heart rot and fruiting bodies of *Ganoderma applanatum*. Other decay fungi observed on bay laurel stumps included *Aleurodiscus aurantius*, *Schizophyllum commune*, and *Trametes versicolor*. Several trees were infected with *Armillaria* spp., which was observed fruiting at the base of the tree (Figure 4).

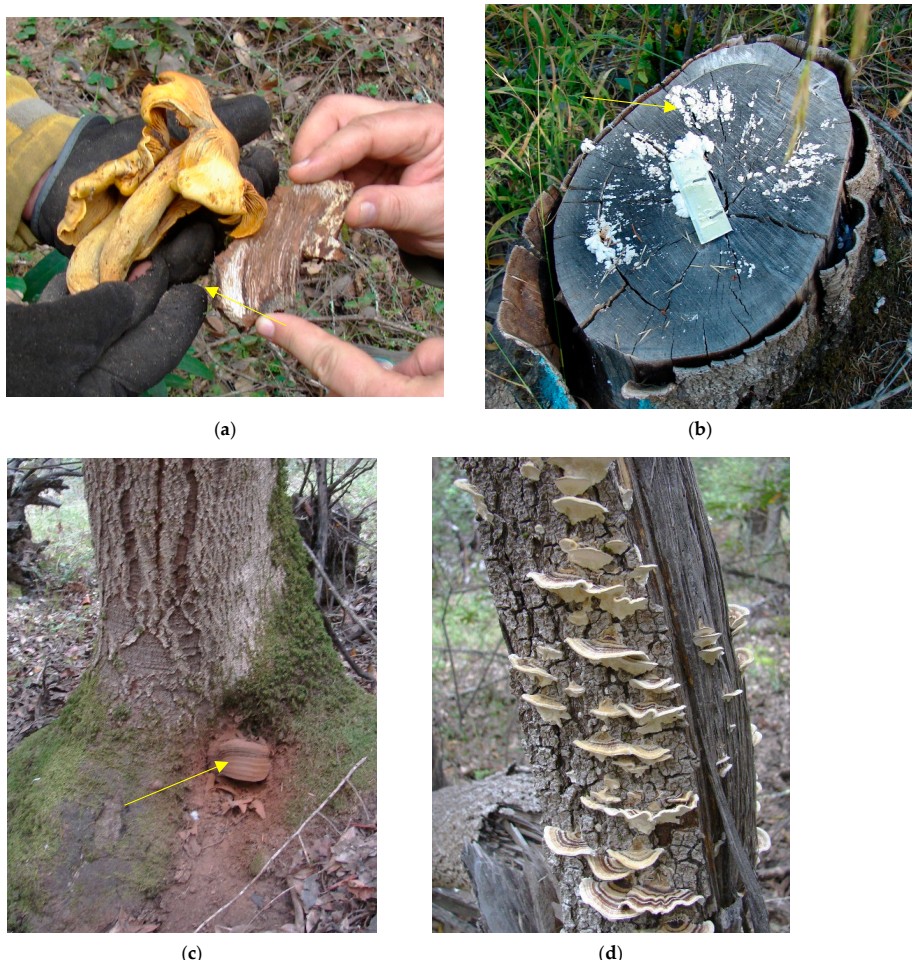

**Figure 4.** Decay fungi on California bay laurel. (**a**) *Armillaria* sp. fruiting body and mycelial fan (arrow) under bark; (**b**), *Aleurodiscus aurantius* (arrow) fruiting on herbicide treated stump; (**c**) *Ganoderma applanatum* fruiting body (arrow); (**d**) *Trametes versicolor* fruiting bodies.

## 2.2. Field Trials

### 2.2.1. Tanoak

The mean stump diameter was 20 cm (range 5–45 cm), and there was no significant difference in the stump diameter among the treatments (Table 1). There was no significant difference in sprouting between stumps treated with the formulation and the formulation plus inoculum, and these were not significantly different from the untreated stumps. The two herbicide treatments had the fewest live sprout clumps. The hack-and-squirt treatment with Arsenal completely inhibited sprouting at all measurement times. Of the biocontrol treatments, sprout clumps on stumps treated with Chontrol™ paste formulations with and without inoculum were smaller in diameter than in those treated with the liquid formulation treatments; therefore. the paste formulation was chosen for further experimentation on bay laurel (Figure 5).

**Table 1.** One-way ANOVA results for variables measured on tanoak stumps one, two, and four years after treatment. Chi-squared and *p*-values are shown for each variable.

| 2010 | Sprouting Treatment |
| --- | --- |
| Live sprouts | 388.95, 0.000 *** |
| Height of tallest sprout | 366.41, 0.000 *** |
| Number of dead sprouts | 65.129, 0.000 *** |
| Deer browse | 117, 0.000 *** |
| Stump diameter | 4.3668, 0.6272 |
| **2011** | **Sprouting Treatment** |
| Live sprouts | 286.97, 0.000 *** |
| Sprout clump diam | 516.51, 0.000 *** |
| Height of tallest sprout | 485.06, 0.000 *** |
| Number of dead sprouts | 114.58, 0.000 *** |
| **2014** | **Sprouting Treatment** |
| Live sprouts | 23.101, 0.001 ** |
| Height of tallest sprout | 24.711, 0.004 ** |
| Number of dead sprouts | 28.818, 0.000 *** |
| Fungi | 20.952, 0.002 ** |

** significant at $p = 0.01$, *** significant at $p = 0.001$.

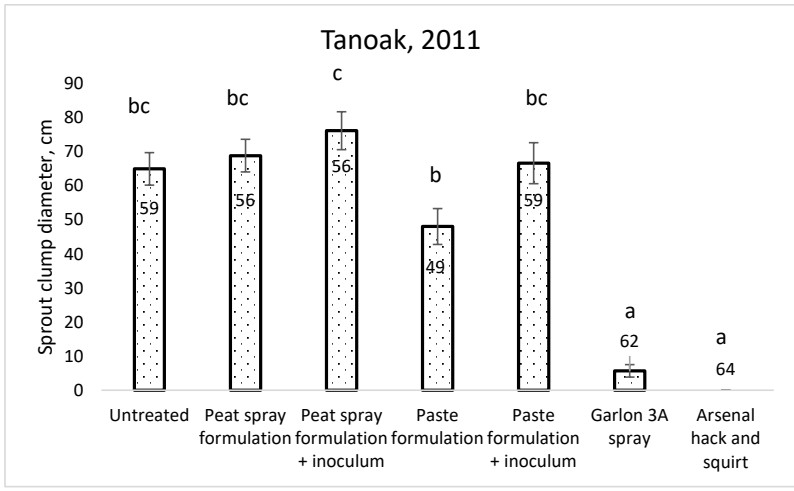

**Figure 5.** Sprout clump diameter on tanoak stumps two years after treatment. Columns with different letters are significantly different at $p = 0.05$, Welch's ANOVA, Games–Howell test.

Analysis of the data showed no differences between the treatments for the number of sprout clumps and sprout height among the biological controls, the formulation-only treatments, and the untreated stumps during the five-year period. Both herbicide treatments were significantly different from the other treatments and had fewer sprout clumps. The most effective treatment after five years was the hack-and-squirt herbicide treatment. Many stumps had decay and fruiting (*Trametes versicolor* and *Lenzites betulina*) with very little resprouting. Herbicide treatments had less fruiting and decay than other treatments. (data not shown).

### 2.2.2. Bay Laurel

The mean stump diameter for bay laurel was 18 cm (range 5–65 cm). There were no significant differences among cutting treatments and stump treatments for diameter. For the cutting treatments, the tall stumps (cut 2×) had more sprouting and deer browse than the cut 1× stumps in 2014. By 2016, there was no difference between the cutting treatments for any variables measured (Table 2).

**Table 2.** Two-way ANOVA results for variables measured on California bay laurel stumps one and three years after treatment. Chi-squared and *p*-values are shown for each variable and interaction term.

| 2014 | Sprouting Treatment | Cutting Treatment | Interaction |
| --- | --- | --- | --- |
| Live sprouts | 30.7640, 0.000 *** | 4.3083, 0.03793 * | 0.0102, 0.99973 |
| Height of tallest sprout | 14.1139, 0.002754 ** | 0.4721, 0.492005 | 0.5370, 0.910691 |
| Number of dead sprouts | 50.294, 0.000 *** | 13.185, 0.000 *** | 1.219, 0.7484413 |
| Deer browse | 77.839, 0.000 *** | 4.841, 0.027789 * | 11.921, 0.007659 ** |
| Stump diameter | 1.14941, 0.7652 | 0.26954, 0.6036 | 1.45238, 0.6933 |
| **2016** | **Sprouting Treatment** | **Cutting Treatment** | **Interaction** |
| Live sprouts | 45.267, 0.000 *** | 1.959, 0.1617 | 5.538, 0.1364 |
| Loose bark | 9.4759, 0.02359 * | 2.0471, 0.15250 | 5.8841, 0.11739 |
| Decay fungi present | 14.2855, 0.002541 ** | 0.0150, 0.902537 | 3.3625, 0.339032 |
| Dead (no sprouting) | 36.978, 0.000 *** | 0.225, 0.6355 | 3.725, 0.2927 |

* significant at *p* = 0.05, ** significant at *p* = 0.01, *** significant at *p* = 0.001.

As with tanoak, herbicide treatment was the most effective against resprouting on bay laurel during the time measured. There was no significant difference between the untreated and formulation or formulation + inoculum treatments for sprouting in 2014 and 2016. However, in 2016 the herbicide and formulation + inoculum treatments were not significantly different from each other (Figure 6a). In 2016, the number of dead stumps for the inoculum treatment was significantly greater than the untreated (Figure 6b). The herbicide-treated stumps had the highest proportion of dead stumps. The fewest decay fungi were observed on the untreated stumps.

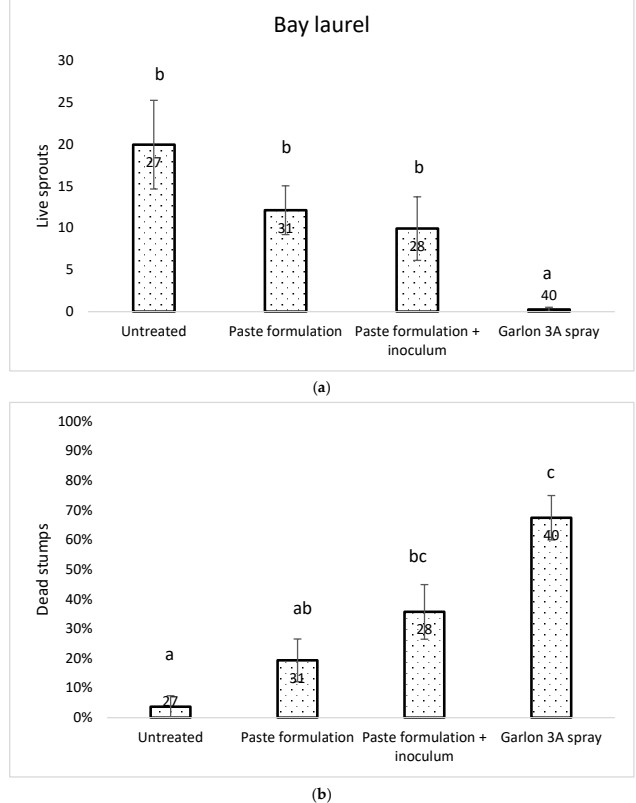

**Figure 6.** Number of live sprouts (**a**) and percent of stumps in each treatment group that were dead (no sprouting, fruiting bodies of decay fungi present) (**b**) of California bay laurel three years after treatment. Columns with different letters are significantly different at *p* = 0.05, Welch's ANOVA, Games–Howell test.

## 3. Discussion

This is the first study to assess the biological control of tanoak and bay laurel with *C. purpureum*. However, *C. purpureum* has been used as a stump-sprouting control agent on other weedy tree species in commercially valuable conifer plantation sites and utility rights-of-way [15,21]. The use of *C. purpureum* as a biological control agent has been investigated in Canada, Belgium, The Netherlands, Lithuania, Finland, New Zealand, and the US. The targeted weedy trees include Red alder (*Alnus rubra*), Paper birch (*Betula papyrifera*), Poplar (*Populus tremuloides*), Black cherry (*Prunus serotina*), Pin cherry (*Prunus pensylvanica*), Black locust (*Robinia pseudoacacia*), Rowan (*Sorbus aucuparia*), Gorse (*Ulex europaeus*), and willow (*Salix* spp.) [14,17,18,22–26]. Application of the formulated *C. purpureum* inoculum on freshly cut stumps leads to tree vessel blockage. Induced dehydration combined with fungal toxins (sterpurens, sesquiterpene metabolites) [27], strengthens the adverse effects of the fungus in preventing the resprouting of stumps. The fungus consumes carbohydrates, and during that process, it decomposes the wood stump with help of a large set of extracellular enzymes [28,29]. The wood stump decay process is a dynamic biological activity, and therefore, treated stumps are not decayed immediately. Depending on the virulence of a selected *C. purpureum* isolate to decay wood and the resistance of the targeted tree species, stump mortality begins as early as two months later and continues for four years after the treatment [16]. In our study, there was some evidence that the formulation (with and without *C. purpureum* inoculum)-treated tanoak stumps supported more decay fungi (including *C. purpureum*) than the untreated controls and the herbicide treated stumps. This is in agreement with the suggestion by Harper et al. [30] that formulation enhances the efficacy of *C. purpureum*. Furthermore, decreased colonization of herbicide-treated stumps relative to the formulation, inoculum, and untreated stumps may be due to a less favorable environment for these fungi. Korzeniewicz et al. [31] found higher levels of colonization by basidiomycete fungi in untreated black cherry stumps larger than 5 cm in diameter than in stumps treated with glyphosate. Our results showed no differences between the treatments for the number of sprout clumps and sprout height among the biological controls, the formulation-only treatments, and the untreated tanoak and bay laurel stumps. The most effective treatment after one year was the hack-and-squirt herbicide treatment. As with tanoak, herbicide treatment was the most effective against resprouting on bay laurel after 1–2 years post-treatment. Previously, in many studies, it was shown that stump treatment with *C. purpureum* against selected tree targets (e.g., alder, paper birch, poplar, and willow) resulted in a significantly higher level of stump mortality compared with controls. Furthermore, in many cases, the efficacy of *C. purpureum* treatment on stump mortality was as high as synthetic chemical herbicides [18,22]. However, such a positive effect of *C. purpureum* was not observed in our tanoak and bay laurel trials after 1–2 years post-treatment. Our findings corroborate the results of Wall [32] and Roy et al. [33], who have drawn similar conclusions on the efficacy of *C. purpureum* on pin cherry (*Prunus pensylvanica*) in Canada and claimed that some hardwood tree species may be completely resistant to this pathogen. Furthermore, Hamberg et al. [25] and Becker et al. [34] found limited effects of *C. purpureum* on sprouts in two different poplar species (*Populus tremula* and *Populus tremuloides*, respectively) one year after treatment. In addition, Menkis et al. [35] found limited or no effects on the mortality of elm (*Ulmus minor*) stumps treated with *C. purpureum* one year post-treatment to control the spread of Dutch elm disease caused by the invasive alien pathogen *Ophiostoma novo-ulmi*. The approach of biological control of weedy tree species using *C. purpureum* or other wood decay fungi does not work as rapidly as chemical herbicide treatments since the decay process and stump mortality begin as early as two months later and continue for four years after the treatment [16]. Overall, many factors may have contributed to the efficacy of *C. purpureum* posttreatment at the tanoak and bay laurel field trials sites:

1. Target tree species: the ability of *C. purpureum* to infect different tree species varies greatly because of the ability of a tree to resist the fungal infection and the physiological state of a tree species. There is evidence that the *C. purpureum* and other decay fungi can inflict the most severe damage to target tree species when the amount of soluble carbohydrates is the highest in wood (i.e., when the tree is growing intensively) [17,36]. Thus, it is important to time *C. purpureum* applications when the resistance of trees against the fungus is lowest for the best sprout control efficacy. Neither tanoak (*Notholithocarpus densiflorus*) nor bay laurel (*Umbellularia californica*) are listed as hosts for *C. purpureum* [19]. This may explain the delay of the response and possible resistance of tanoak and bay laurel to *C. purpureum* infection. Similar findings were observed on black locust (*Robinia pseudoacacia*) and sea buckthorn (*Hippophae rhamnoides*) because of either resistance of these trees to *C. purpureum* or a significantly delayed response to fungal infection [17];

2. Virulence of selected *C. purpureum* isolates: The efficacy of *C. purpureum* isolates varies, with some isolates more virulent in preventing sprouting than others [22,37,38]. In laboratory conditions, the laccase manganese peroxidase enzyme production of *C. purpureum* isolates has been shown to correlate with birch sprout control efficacy in field conditions [39]. In addition, crossbreeding of *C. purpureum* has been performed to increase biocontrol efficiency [40]. In our field trial at the tanoak site, we used isolate *C. purpureum* PFC2139, which was selected on the basis of its virulence, expressed as canker size on red alder (*Alnus rubra*) seedlings in greenhouse conditions [37]. The selection of *C. purpureum* isolate PFC2139 in our tanoak field trials is not necessarily "ecologically fit" to control tanoak resprouts; hence, there is a potential to explore the use of native decay fungi occurring in southwest Oregon forests. This includes using *Stereum hirsutum*, closely related to *C. purpureum*. This fungus has already been shown to be an aggressive and common colonizer of wounds in hardwood tree species. In addition, *S. hirsutum* was also found in association with an unusual decline of tanoak sprouts [41], and therefore, it may possess the potential to control stump sprouting in tanoak. In the case of bay laurel field trials, the *C. purpureum* isolate was selected on the basis of the bay laurel stem inoculations under greenhouse conditions at the Canadian Forest Service, Pacific Forestry Centre. On the basis of the inoculation results, we selected isolate PFC2249 for further formulation using the same technology used for *C. purpureum* isolate PFC2139 in tanoak trials. As with tanoak trials, the *C. purpureum* isolate 2249 that was isolated from a canker on *Prunus* spp. for use in our bay laurel field trials is not necessarily "ecologically fit" to control bay laurel. This may explain the delay to the response or possible resistance of bay laurel to *C. purpureum* isolate PFC2249 infection and control of bay laurel resprouts; hence, there is a potential to explore the use of native decay fungi occurring in northern California forests. This includes using the most common decay fungi *Ganoderma applanatum*, *Schizophyllum commune*, and *Trametes versicolor* naturally occurring in bay laurel trials.

3. Environmental conditions: In laboratory conditions, the optimum growth temperature for *C. purpureum* is 24–25 °C, but in extreme temperatures (e.g., 0 °C and over 35 °C), growth is severely inhibited [42]. However, in the field conditions, *C. purpureum* is able to withstand high temperatures (30–40 °C) [43]. Because of logistical reasons, we have treated tanoak and bay laurel stumps with *C. purpureum* in our field trials during the summer (temperatures were as high as 35 °C); hence, there was a delay in sprout control in the first 1–2 years post-treatment. Our results corroborate those of Hamberg and Hantula [44], who found a similar delay in the sprout efficacy control of *C. purpureum* (i.e., the growth of mycelia within the stump).

## 4. Materials and Methods

### *4.1. Laboratory*

#### 4.1.1. Bay Laurel Isolate Selection

Three isolates of *Chondrostereum purpureum* were selected from the University of California, Riverside, culture collection to be utilized as potential biocontrol agents for California bay laurel (*Umbellularia californica*) [45]. These isolates (2249, 2367, and 2434) were isolated from *Prunus dulcis*, *Prunus persica*, and *Prunus dulcis*, respectively (Table 3). Screening of *C. purpureum* isolates was performed in the laboratory and the Pacific Forestry Centre greenhouse. Isolates were grown in Petri dishes on potato dextrose agar media (PDA). Detached bay laurel stems were cut and trimmed from all leaves to a size of 8 centimeters. In the greenhouse, the stems of potted bay laurel were used. The stems were inoculated with the test isolates or a blank PDA disk. and the inoculum site was sealed with parafilm. The isolates were re-isolated from the symptomatic areas of the bay laurel stems and re-inoculated onto bay laurel three times and re-isolated for subsequent inoculations.

**Table 3.** Isolates of *Chondrostereum purpureum* tested for biocontrol activity against California bay laurel (*Umbellularia californica*).

| Isolate ID | Year Isolated | Host | Location | Collector | Identifier |
|---|---|---|---|---|---|
| PFC2249 | 2001 | *Prunus dulcis* | Modesto CA | J. E. Adaskaveg | J. E. Adaskaveg |
| PFC2367 | 2001 | *Prunus persica* | Parlier CA | T. Michailides | J. E. Adaskaveg |
| PFC2434 | 2002 | *Prunus dulcis* | Modesto CA | J. E. Adaskaveg | J. E. Adaskaveg |

#### 4.1.2. Identification of Fungi Collected from Stumps

Fungi on tanoak and bay laurel collected from the Oregon and California field sites were identified on the basis of the morphology of the fruiting body or isolation into pure culture. The identities of the fungi were verified by sequence analysis of the ITS rDNA. DNA was extracted from fruiting bodies and pure cultures using the Qiagen DNeasy Plant Mini Kit and the ITS region of rDNA was amplified with primers ITS6 and ITS4 [46]. Excess nucleotides and primers were removed using the ExoSAP-IT reagent (Affymetrix) following the manufacturer's recommendations. The PCR product was cycle-sequenced to incorporate fluorescent dye terminator labels and electrophoresed and analyzed on an Applied Biosystems 3730xl capillary electrophoresis instrument. The identification of isolates by sequence was determined using alignments with voucher sequences from the GenBank database and comparing isolate sequences with those of closely related species using MEGA v. 10.1.8 [47].

### *4.2. Field Trials*

#### 4.2.1. Tanoak

A site was selected on private timberland near Brookings, OR (42.099, −124.286, 386 m elev.) in September 2009. Overstory vegetation on the site was composed of Douglas fir and tanoak with an understory of many hosts for *P. ramorum*, including salal, Oregon grape, Pacific rhododendron, and evergreen huckleberry.

A total of 420 tanoak trees ranging from 5 cm to 45 cm dbh (diameter at breast height) were selected and cut, leaving stumps approximately 40 cm tall. A total of 7 treatments (Table 4) were applied to the stumps in November 2009. Treatments were applied in 3 blocks of approximately 20 stumps for each treatment. Blocks were randomly arranged within the plot.

**Table 4.** Treatments applied to tanoak stumps in 2009. Treatment efficacy against resprouting was evaluated in September 2010 and 2011 and May 2014.

| Treatment | Description |
| --- | --- |
| Control | No treatment. |
| ChontrolTM liquid w/ inoculum [a] | Peat spray formulation containing Chondrostereum purpureum isolate PFC2139 105 to 107 Colony Forming Units (CFU) per L. |
| ChontrolTM liquid w/o inoculum | Peat spray formulation only. |
| ChontrolTM paste w/ inoculum | Paste formulation containing Chondrostereum purpureum isolate PFC2139 $1 \times 102$ CFU per gram. |
| ChontrolTM paste w/o inoculum | Paste formulation only. |
| Garlon 3A [b] | Apply triclopyr (Garlon 3A (Amine)), cut 50–50 with water, plus dye to all exposed cambium immediately after cutting (within 30 min). Exposed cambium includes the stump surface and bark tears that occurred during falling. |
| Hack and squirt [c] | Inject imazapyr (Arsenal®) cut 50–50 with water, 1 hack (1 mL solution/hack) per 3 inches diameter) plus dye using the hack-and-squirt method. Hacks were made at or below stump height (40 cm). |

[a] Chontrol™ previously produced by Mycologic, Inc., c/o IDC, The University of Victoria, Victoria, BC, Canada V8W 2Y2. EPA Reg. No. 74200-2, EPA Est. No. 074200-CAN-001. Current registration is LALCIDE CHONDRO produced by Danstar Ferment AG, Switzerland. Canadian representative Lallemand Inc./LALLEMAND PLANT CARE, 59 Industrial Park Crescent, Unit 1, Sault Ste, Marie, ON, P6B 5P3. [b] Garlon 3A Ultra Herbicide produced by Dow AgroSciences LLC, 9330 Zionsville Rd., Indianapolis, IN, 46268, USA. EPA Reg. No. 62719-37. [c] Arsenal Herbicide produced by BASF Corporation, 26 Davis Drive, Research Triangle Park, NC, 27709, USA. EPA Reg. No. 241-346.

The stump diameter, number of live sprout clumps, number of dead shoots, and height of tallest shoot were measured on treated tanoak stumps in September 2010 and 2011. Other notes were taken on fruiting of *C. purpureum* and other fungi and deer browse. In addition to these measurements, the diameter of each sprout clump was measured in 2011. In 2014, the site was clearcut on the east side, and many stumps were destroyed or buried under slash. Data were collected on the 70 remaining stumps in May 2014.

4.2.2. Bay Laurel

The bay laurel site was identified in Soquel Demonstration State Forest (37.086, −121.905, 211 m elev.) in May 2012. The site has an extensive overstory of bay laurel trees infected with *P. ramorum*. Most of the tanoak had been killed by *P. ramorum* and was replaced with bay laurel. As a result, many of the regenerating Douglas fir beneath these trees showed signs of infection. Other trees on the site included coast redwood, coast live oak, and Pacific madrone. Understory vegetation was sparse because of the dryness of the site and included regenerating Douglas fir, grasses, and herbaceous plants.

Three hundred bay laurel trees at Soquel Demonstration State Forest were selected, mapped, and labeled for treatments in February 2013 (Figure 7). Because of the inability to apply treatments in March because of permit issues, half of the bay laurel trees were cut to 1 m in height in March 2013. All selected bay laurel stems and tall stumps within the treatment plots were manually brushed using chainsaws to a stump height of approximately 30 cm prior to treatment application in July 2013. The two cutting treatments allowed us to examine whether there were differences in treatment effectiveness on stumps that had been cut and re-cut prior to treatment application versus being cut one time and may provide more options to forest managers when planning treatments. Stem treatments included a paste formulation of *C. purpureum* PFC2139 (commercial product Chontrol; currently commercialized as LALCIDE CHONDRO, Lallemand Inc., Sault Ste. Marie, ON, Canada), formulation only, cut stump with no treatment, and spray application of Garlon 4 Ultra herbicide (Table 5).

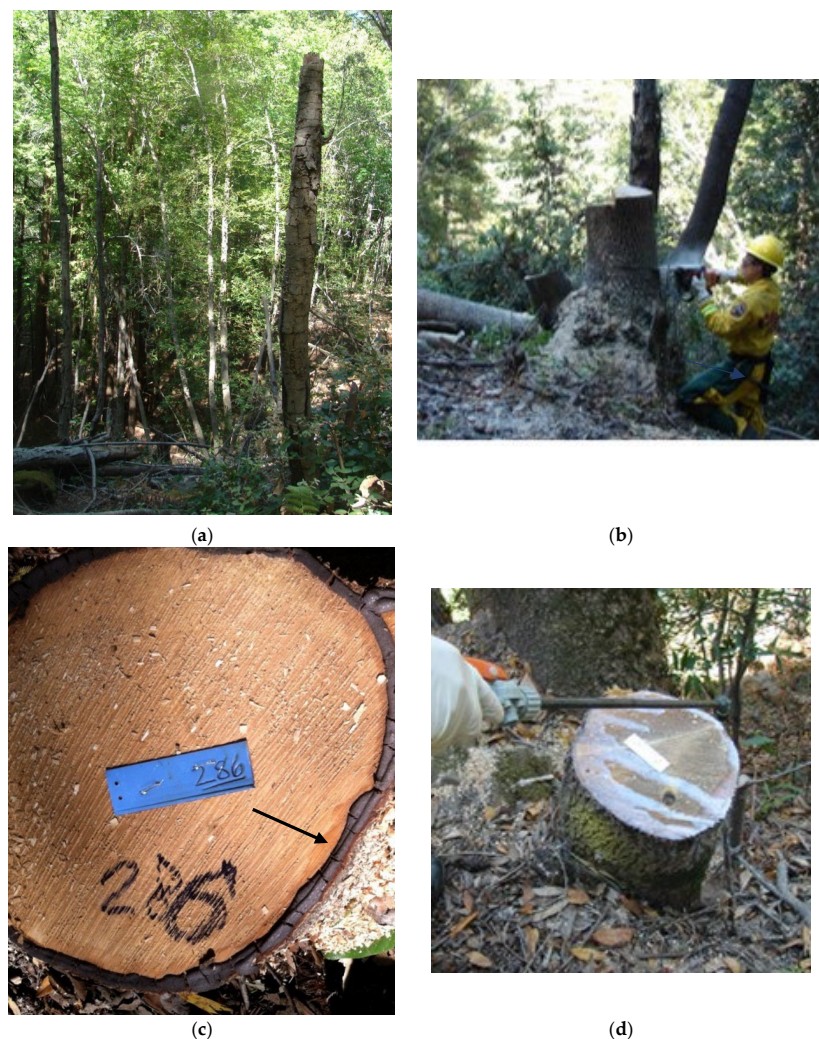

**Figure 7.** Soquel Demonstration State Forest site and treatments: (**a**) Pre-treatment site showing dead tanoak with bay laurel overstory; (**b**) cutting a tall stump; (**c**) stump with Chontrol paste on edge of stump (arrow) after application (photo: Nathan Stacey); (**d**) applying herbicide spray to cut stump.

**Table 5.** Treatments applied to cut bay laurel stumps and girdled trees in July 2013; treatment efficacy against resprouting was evaluated in May 2014 and June 2016.

| Treatment | Description |
|---|---|
| Control | No treatment. |
| Chontrol™ paste w/inoculum [a] | Paste formulation containing *Chondrostereum purpureum* isolate PFC2249 $1 \times 10^2$ CFU per gram. |
| Chontrol™ paste w/o inoculum | Paste formulation only. |
| Garlon 4 Ultra [b] | Apply triclopyr (Garlon 4 Ultra (Amine)), 30% in oil, diesel fuel, or kerosene, plus dye to all exposed cambium immediately after cutting (within 30 min). Exposed cambium includes the stump surface and bark tears that occurred during falling and the exposed sapwood area on trees with the girdling treatment. |

[a] Chontrol™ previously produced by Mycologic, Inc., c/o IDC, The University of Victoria, Victoria, BC, Canada V8W 2Y2. EPA Reg. No. 74200-2, EPA Est. No. 074200-CAN-001. Current registration is LALCIDE CHONDRO, produced by Danstar Ferment AG, Switzerland. Canadian representative Lallemand Inc./LALLEMAND PLANT CARE, 59 Industrial Park Crescent, Unit 1, Sault Ste, Marie, ON, P6B 5P3. [b] Garlon 4 Ultra Herbicide produced by Dow AgroSciences LLC, 9330 Zionsville Rd., Indianapolis, IN, 46268, USA. EPA Reg. No. 62719-37.

The number of live sprout clumps, number of sprout clumps dead or with dieback, the height of the tallest sprout, and stump diameter per stump for each treatment were assessed in May 2014 after the first growing season post-treatment. Deer browsing of sprouts was noted for each stump. Three years after treatment (June 2016), live sprout clumps and whether the stumps had loose bark, decay fungi present, or were dead (no sprouting) were noted.

*4.3. Data Analysis*

Stump sprouting data for each year was analyzed separately. For all data, homogeneity of variance was evaluated using the Fligner–Killeen test. and normality was tested with the Shapiro–Wilk test. General Linear Models (GLMs) were created for each variable followed by ANOVA with standard errors corrected for heteroscedasticity [48]. For tanoak, one-way analysis of variance (ANOVA) was performed followed by Tukey's HSD test if the means were significantly different at $p = 0.05$, with stump sprouting treatments as the factor. For bay laurel, two-factor ANOVA with interactions was performed. The factors for main effects were cutting treatment and stump sprouting treatment. Since the cutting treatments were not significantly different, one-way Welch's ANOVA was performed on stump treatments, followed by the Games–Howell test if means were significantly different at $p = 0.05$. Data analysis was performed in R version 4.1.2 [49].

**5. Conclusions**

This research is the first study to assess the biological control of tanoak and bay laurel resprouts with *C. purpureum*. Our results indicate low efficacy of *C. purpureum* 1–2 years post-treatment when compared with chemical herbicides. Neither tanoak (*Notholithocarpus densiflorus*) nor bay laurel (*Umbellularia californica*) is listed as a host for *C. purpureum*. This may explain the delay in response of tanoak and bay laurel to *C. purpureum* infection. In addition, other factors may be attributed to the low efficacy of *C. purpureum* post-treatment in the tanoak and bay laurel field trials. These factors include (1) resistance of tanoak and bay laurel to infection by *C. purpureum*; (2) the virulence of *C. purpureum* isolates PFC2139 and PFC2249 on inoculated tanoak and bay laurel, respectively, could have not been aggressive enough to inhibit resprouting of these hosts; and (3) environmental conditions, such as high temperatures (up to 35 °C), may have contributed to the slow growth of *C. purpureum* mycelia within the stump, leading to the low efficacy of *C. purpureum*. The biological control approach to resprouting of trees using *C. purpureum* or other wood decay fungi does not work as rapidly as chemical herbicide treatments since the decay process and stump mortality can begin as early as 2 months and continue for four years after treatment. There is an urgent need to conduct long-term monitoring of the field trials to assess the effects of *C. purpureum* on resprout control in tanoak and bay laurel after 3–4 years post-treatment. For future work, we propose using an indigenous isolate of *C. purpureum* selected from tanoak and bay laurel that is adapted to local conditions, as well as other wood decay fungi as formulated product(s), and testing their field efficacy on tanoak and bay laurel.

**Author Contributions:** Conceptualization, S.F.S. and M.E.; laboratory trials, S.F.S.; field trials S.F.S. and M.E.; data analysis, M.E.; writing—original draft preparation, S.F.S. and M.E.; writing—review and editing, S.F.S. and M.E.; supervision, M.E.; project administration, M.E.; funding acquisition, S.F.S. and M.E. All authors have read and agreed to the published version of the manuscript.

**Funding:** This research was funded by USDA Forest Service, State and Private Forestry grant #09-DG-11052021, and the USDA National Institute of Food and Agriculture, McIntire-Stennis Project 1019284.

**Institutional Review Board Statement:** Not applicable.

**Informed Consent Statement:** Not applicable.

**Data Availability Statement:** Data from this project can be requested from the authors.



**Acknowledgments:** We extend our thanks to Ed Orre, Angela Bernheisel, Ellen Goheen, Alan Kanaskie, Nathan Stacey, Gary Chastagner, Grace Sumampong, and Robert Kowbel for their technical support in laboratory and fieldwork conditions. We acknowledge the support of the Canadian Forest Service Pest Risk Assessment Research Program and Washington State University. In addition, we thank Paul de la Bastide (MycoLogic Inc., University of Victoria, BC, Canada) for formulating *Chondrostereum purpureum* isolates for field trial applications.

**Conflicts of Interest:** The authors declare no conflict of interest. The funders had no role in the design of the study; in the collection, analyses, or interpretation of data; in the writing of the manuscript; or in the decision to publish the results.

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
