# Peer review of "Comparative Efficacy of Chondrosterum purpureum and Chemical Herbicides for Control of Resprouts in Tanoak and Bay Laurel"

_pathogens, doi:10.3390/pathogens11050485_

Round 1

Reviewer 1 Report

The manuscript by Shamoun and Elliott describes tests of the efficacy of the sap-rotting fungus Chondrosterium purpureum in preventing resprouting of two trees that are hosts of the destructive forest pathogen Phytophthora ramorum - tanoak and California bay laurel - in forests in Oregon and California, USA. The paper presents only negative results — in all cases, the biological control does not control resprouting significantly better than identical treatments without the fungal inoculum or, by most measures, than untreated controls. However, I think there is a case to be made for describing what doesn’t work — and potentially, why — when it comes to developing biological control strategies and comparing them to chemical alternatives. 

Below are several points that I think should be addressed by the authors:

  • The rationale provided for controlling resprouting of tanoak is that sprouts act as sources of inoculum; however, to my knowledge, tanoak is considered a dead-end host. I think this point should be clarified and, in the absence of evidence for transmissive potential of tanoak, the rationale for including tanoak control in the study needs to be clarified. 
  • The labeling of Figure 5 is unclear. The y-axis label notes the number of dead sprouts — should this be live sprouts instead?
  • Figure 6(a) indicates (by the letters above each bar) that the paste formulation + inoculum treatment does not significantly differ from the Garlon 3A spray treatment, but a visual comparison of the bars and associated error bars don’t appear consistent with that conclusion. I would recommend the following with all of the barplots and analyses: (1) confirm that the normality and homogeneity of variance assumptions of ANOVA are met by the data; (2) plot individual data points along with the bars; (3) provide sample numbers for each treatment in the figure captions; provide the Tukey HSD p-value for the contrast between the paste formulation + inoculum treatment and the Garlon 3A spray treatment.
  • Table 3 seems unnecessary, as the isolate numbers and hosts are already provided in the text, and the year collected could be easily added to that information. 
  • The point made in lines 136-137 about observing less fungal fruiting and decay in herbicide treatments should receive some attention in the Discussion, as it seems like there are some important ecological implications of using treatments that inhibit natural decay processes.
  • The conclusions (lines 375-380) are quite brief and fail to discuss the important point that the C. purpureum treatments were not effective — on tanoak, there was no improvement of results at all, and in bay, the C. purpureum treatment was worse than the chemical controls and offered no increase in efficacy over identical treatments but without the inoculum. 

In addition, below are some specific suggestions regarding the text itself:

  • Lines 30-32: Keywords “Biological control”, “Chondrostereum purpureum”, “Phytophthora ramorum”, “California bay laurel”, and “tanoak” already appear in the title and/or abstract, so please confirm that these are appropriate keywords as per the journal’s instructions to authors.
  • Lines 34-36: The wording of this sentence is a bit awkward.
  • Line 36: replace “The epidemics of an invasive…” with “Epidemics of the invasive…”
  • Line 38: replace “in nursery trade” with “in the nursery trade”; replace “in the west coast” with “on the west coast”
  • Lines 45-48: Some of the text is redundant. Consider replacing “In southwestern Oregon, long-45 term efforts have been made to eradicate P. ramorum. An aggressive eradication program included use of chemical herbicides, cutting, and burning of tanoak in an effort to eradicate P. ramorum “  with “In southwestern Oregon, an aggressive eradication program has included use of chemical herbicides, cutting, and burning of tanoak in an effort to eradicate P. ramorum.”
  • Lines 56-57: As mentioned above, I’m not aware of evidence that tanoak is a transmissive host.
  • Line 61: replace “recreation use” with “recreational use sites”
  • Line 62: replace “cultural values” with “cultural value” 
  • Line 62-65: There is some repetition of ideas here; these two sentences can be condensed.
  • Line 79: Phytophthora ramorum should be italicized in the Figure 1 caption.
  • Line 118: there is an extra period before the semicolon in point (c) of the Figure 4 caption.
  • Line 162: the second “(a)” in the Figure 6 caption should be “(b)”.
  • Line 175: change “stump” to “stumps”
  • Line 179: change “help of large set” to “help of a large set”; change “the decay of wood stump process” to “the wood stump decay process”
  • Line 194: change “resulted in significantly higher” to “resulted in a significantly higher”
  • Line 197: change “such positive effect” to “such a positive effect”
  • Lines 198-199: change “corroborate with those investigations of” to “corroborate the results of”
  • Lines 199, 202, 204, 261, and perhaps elsewhere: The years are not necessary since the citation number is given after the author names.
  • Lines 207-208: replace “Biological control of weedy tree species approach using” with “The approach of biological control of woody tree species using”
  • Line 209: replace “since decay process” with “since the decay process”
  • Line 225: replace “either of resistance” with “either resistance” 
  • Line 226: replace “fungus” with “fungal”
  • Line 232: replace “we have used” with “we used”
  • Line 233: replace “on basis” with “on the basis”
  • Line 239: replace “shown to be aggressive” with “shown to be an aggressive”
  • Line 242: replace “in case of” with “in the case of”
  • Line 244: there is an extra period at the end of the sentence that ends in Centre 
  • Line 256: replace “the growth is” with “growth is”
  • Lines 260-261: replace “Our results corroborate with Hamberg and Hantula” with “Our results corroborate those of Hamberg and Hantula”
  • Line 271: replace “potato-dextrose-agar” with “potato dextrose agar”
  • Line 301: should “diameter” be “dbh”?
  • Line 322-323: include commas after “measurements” and “2014”
  • Line 427: Salix should be italicized
  • Lines 448-449: “Colletotrichum clavatum” should be italicized;  “Sp.” and “Nov.” should not be capitalized; there is an unnecessary quotation mark after “Italy” 
  • Line 455: replace “deployment Chondrostereum” with “deployment of Chondrostereum

Reviewer 2 Report

I have carefully read MS which was submitted for consideration in the Pathogens (MDPI). This is an interesting and valuable case study providing novel information about the occurrence and method of biological control of a highly destructive plant pathogen Phytophthora ramorum. The disease kills oak and other species of trees and has had devastating effects on the oak populations in California and Oregon, as well as being present in Europe. Symptoms include bleeding cankers on the tree's trunk and dieback of the foliage, in many cases leading to the death of the tree. The taxon was first discovered in California in 1995 when large numbers of tanoaks (Notholithocarpus densiflorus) died mysteriously.

P. ramorum is a relatively new disease, and several debates have occurred about where it may have originated or how it evolved. One of the major reasons that identifying the natural range of this organism is difficult is that it typically will not cause symptomatic or infectious disease in hosts that are adapted to live in concert with it naturally. It is only when this organism leaks into vulnerable habitats with less resistant host species that a notable amount of destruction occurs.

In recent years, intensive research has been carried out on replacing the use of mycoherbicides with biocontrol methods, including the use of the fungal plant pathogen Chondrostereum purpureum.

The objectives of this article were to establish field trials in southwestern Oregon and the Soquel Demonstration State Forest site in California to examine and evaluate the efficacy of the potential application of formulated product of C. purpureum to inhibit the resprouts in Notholithocarpus densiflorus and Umbellularia californica as compared to manual brushing and chemical herbicide treatments. In this paper, the authors show, that the treatment of stumps with the C. purpureum is an effective tool for suppression of resprouting on several species, e.g. Alnus rubra. It is worth noting that this is the first study to assess the biological control of tanoak and bay laurel with C. purpureum.

The paper is in general well written, logically structured, well-illustrated and easy to understand. It also addresses a subject that is of great interest in the scientific community. I suggest changing the title of the article. The abstract is well written. It encapsulates the entire study (a bit of introduction, aim, result and outcome). The introduction is well written as it gives a good background of the research in question. Also, the aim of the study is evident in the beginning and concluding parts. I believe that the Materials and Methods section is well structured and scientifically sound. The results are well presented, figures and tables are correct. Literature reviews in the discussion section of the manuscript are very good. My comments mostly relate to relatively minor issues of interpretation and writing. These comments do not influence a positive impression from the article.

Suggestions:

Title: In my opinion, the title of the article should be changed.

Please correct the formatting of the tables, in the current version of the manuscript does not follow the MDPI recommendations.

Please consider reducing the number of subheadings - are they all needed? For example, 4.2.2. Bay laurel, 4.2.2.1. Site information - isn't it enough X. Study Populations?

L375: Chapter 5. Conclusions should be changed and supplemented, it does not contain conclusions resulting from the article, but only the authors' suggestions on what should be done in the future.

Reviewer 3 Report

The manuscript "Biological control of resprouting in Tanoak and Bay laurel using the fungus Chondrostereum purpureum" is a paper about the bioherbicide for suppression of resprouting on Tanoak and Bay Laurel. It is low novelty in the field of pathogens and diseases because C. purpureum was already used as bioherbicide for many weedy trees but highly practical in the forest management. The ms seems to be appropriate for other journal such as "Forests" on MDPI. Science and logics were good and well-worth publishing without followings.

  1. Authors should improve all low quality photos and their layout. Also, add "arrows" on some figures for indicating each explanation. For exam, in Fig 1(b), which is the lesions of P. ramorum?,  in fig3, where is Chondrostereum?  and in fig 4, mycelial fan under bark could be recognized, so on.
  2. The explanation of figure 6 (b) was absent.
  3. Authors selected the isolate 2249 for the field trials, but actually used the isolate PFC2139. What is "PFC" and this isolate? 
  4. In table 3, authors should provide more details of each isolate, for exam, collected locality, collector, host and so on.

Round 2

Reviewer 3 Report

The ms was well-improved and well-worth-publishing.